# Effect of Process Parameters on the Surface Microgeometry of a Ti6Al4V Alloy Manufactured by Laser Powder Bed Fusion: 3D vs. 2D Characterization

Alberto Molinari [1], Simone Ancellotti [1], Vigilio Fontanari [1], Erica Iacob [2], Valerio Luchin [3], Gianluca Zappini [3] and Matteo Benedetti [1,*]

[1] Department of Industrial Engineering, University of Trento, Via Sommarive 9, 38123 Trento, Italy; alberto.molinari@unitn.it (A.M.); ancellotti.simone89@gmail.com (S.A.); vigilio.fontanari@unitn.it (V.F.)
[2] Fondazione Bruno Kessler (FBK), Via Sommarive 18, 38123 Trento, Italy; iacob@fbk.eu
[3] Lincotek Additive, Via Al Dos de la Roda 60, 38057 Trento, Italy; valerio.luchin@lincotek.com (V.L.); gianluca.zappini@lincotek.com (G.Z.)
* Correspondence: matteo.benedetti@unitn.it; Tel.: +39-0461-282441

**Abstract:** The influence of the main process parameters, laser power, point distance and time exposure, on the surface microgeometry of Ti6Al4V specimens produced by a pulsed powder bed fusion process was investigated. A 3D characterization was carried out and collected data were elaborated to reconstruct the surface and to determine both the 3D and the 2D material ratio curves along different directions. The 3D material ratio curve gives a slightly lower material ratio of peak zone Mr1 and higher material ratio of valley zone Mr2, reduced peak height Rpk and reduced valley height Rvk than the 2D curves. Roughness is greater in the 3D analysis than in the 2D one, skewness is the same and kurtosis increases from <3 in 2D to >3 in 3D. Roughness and skewness increase on increasing point distance and decreasing time exposure and laser power. Within the investigated ranges (27.3–71.2 J/mm$^3$), an increase in energy density reduces the surface roughness while skewness and kurtosis are not significantly affected. The results indicate that a 3D approach allows better characterization of the surface microgeometry than a 2D one.

**Keywords:** laser powder bed fusion; surface microgeometry; roughness; skewness; kurtosis; material ratio



## 1. Introduction

Additive manufacturing (AM) of Ti and its alloys finds broad application especially in the biomedical field, in which surface roughness impacts on properties [1]. The surface microgeometry of parts produced by AM is fundamental to determine the quality of the products. It plays an important role not only in the final properties of the parts, but also in their final cost and, ultimately, the cost competitiveness of the technology.

In laser powder bed fusion (LPBF) processes, asperities of solidified layers in the surface parallel to the building plane may collide with the recoating blade causing production interruption [2] or the formation of pores that impair the structural integrity of the final products [3]. Fatigue strength is not only influenced by residual stress and porosity, but also by surface roughness. Surface defects act as stress raisers, thus surface finishing aimed at reducing surface roughness extends the fatigue life, as experimentally proved in [4,5]. Chen et al. [6] noted that an appropriate contour scan strategy could improve the surface roughness impact on fatigue-life of LPBF Ti6Al4V alloy samples. Thus, optimization of LPBF process parameters and strategy, along with post finishing, must be addressed also in view of roughness reduction. This is particularly important if specific tribological properties are pursued, as surface roughness affects friction and wear. As an example, the surface of the knee prosthesis in contact with the UHMW polyethylene spacer and kneecap must be perfectly finished to avoid abrasion of the counteracting polymer. Furthermore,

surface roughness affects the aesthetic quality of parts, and this is a fundamental concern in the jewelry industry where AM techniques are finding increasing application in the last decade [7], and in the fashion industry that is looking at AM with a growing interest. In all these cases, as-built parts cannot be used without grinding and/or polishing operations. When mechanical or electro finishing is necessary, the worse the as-built surface microgeometry, the more material that must be removed and this can impact negatively on the costs related to finishing: the reuse of machining chips represents an additional cost as they must be cleaned from the machining lubricants and their formation represents a waste of raw material, impairing the economic competitiveness of the additive technology. Even after finishing, visible traces of the previous as-build morphology remain on the parts [8].

A proper characterization of the surface microgeometry of metal additive manufactured components necessitates a sophisticated surface metrology. This is not an easy task, as severe irregularities are present at different length scales: discontinuities, vertical walls and re-entrant features. Furthermore, a typical surface shows irregular features hard to capture as specific patterns, balling, spatters, lose or partially melted particles. The top surface of a layer-by-layer build part is influenced by the texture of the previous layer underneath, producing different surface features at multiple length scales (wavelengths). In view of these complexities, measurement strategy, setting and technology should be tailored to the specific surface features.

Problems arise in measuring LPBF parts' surfaces both with mechanical and optical procedures. Profile measurements via a stylus-based contact instrument are the most used in the industry as they are economic and do not require a high level of training. However, for complex topography, profile-based measurements as well as texture parameters cannot provide exhaustive information. Too steep sides of surface asperities or deep valleys may cause stylus jamming and discontinuous contact during the scan and even the stylus damage. Furthermore, soft materials can be damaged or worn under the passage of the stylus, and thus contact radii and force must be carefully chosen [9]. As a consequence, contact measurements based on a traveling stylus may not capture completely the real surface morphology. Stylus tip and cone angle should be chosen to prevent damage, when passing over tall, steep protuberances or deep craters [9]. According to Cabanettes et al. [8], contact stylus measurements are not suited for AM surfaces. On the contrary, non-contact techniques are less risky, though their accuracy is affected by reflective properties of the material, which are likely to be non-uniform. X-ray computed tomography (XCT) seems the most promising despite the limited spatial resolution [9].

Surface microgeometry characterization consists of acquiring topography information from profile measurements and, successively, of extrapolating numbers or quantities that are indicative of the relevant aspect of the texture. ISO 4287 [10] illustrates terms, definitions, and parameters for the determination of surface microgeometry (roughness, waviness and primary profile) obtained from profiling methods. The material ratio curve, also known as the Abbott–Firestone curve, has been used for texture analyses [11]. However, due to the texture of the surfaces manufactured by PBF technologies, a 3D characterization and the resulting areal parameters defined by ISO25178-2 [12] may be more suitable to characterize their microgeometry [8]. Senin [13] developed an innovative algorithm to automatically identify and characterize relevant topographic formations of LPBF printed surfaces, such as spatters, welding tracks and ripples, and to quantify their geometrical properties.

The characterization of the surface microgeometry of additively manufactured metallic parts is the subject of several studies in literature. Cabanettes et al. [8] investigated the surface microgeometry of Ti6Al4V samples fabricated by LPBF with different inclination. At micrometer scale, morphology of the welded tracks reflects the generated surfaces. The more inclined are the surfaces, the bigger is the amount of partly melted particles and, thus, the more fractal becomes the surface microgeometry. Yang et al. [14] studied the effect of linear energy density (LED) on the surface roughness of the vertical planes. Increasing the energy density, the stability of weld tracks improves up to a certain point, above which the trend is reversed, as was clearly described by Yadroitsev [15]. The higher

the stability of weld tracks, the lower the roughness of vertical surfaces. Gu et al. [16] reported that stable molten pool and smooth track surface can be obtained by improving surface tension and wettability. This can be achieved with higher temperatures reached at a high energy density. Wang et al. [17] carried out empirical investigations on how LED affects the roughness and densification behaviour of AlSi10Mg samples. By increasing LED, density increases and roughness decrease. However, excessive LED decreases the surface quality due to liquid instability, along with defects formation. Jozwik et al. [18] investigated the relationship between the laser power and morphological characteristics of the surfaces. The basic parameters' average height (Sa) and root-mean-square height (Sq) do not show a linear dependence with laser power. In addition, the authors pointed out kurtosis (Sku) and skewness (Ssk) as better descriptors of the surface morphologies obtained by LPBF. Hitzler et al. [19] carried out an experimental campaign aimed at investigating the surface roughness dependencies in plane. Roughness of side faces perpendicular to the layers seems not to be influenced by the irradiation sequence and inert gas stream, but rather by irradiation setting, i.e., the higher is the energy density the rougher are all side faces. Conversely, an increment of energy density leads to a better surface quality with reduced roughness. Recently, Weißmann et al. [20] investigated the impact of build orientation on surface morphology of 3D printed Ti6Al4V specimens, laying the basis to optimized roughness for medical applications. The influence of the orientation was investigated by Leary [21] who highlighted that staircase effects and adhered particles determine roughness when inclination tends towards horizontal and vertical, respectively.

Spierings et al. [22] investigated the influence of the particle size distribution on Ra of an AISI316L stainless steel manufactured by LPBF and demonstrated that the processing of powders with finer particles results in a lower roughness, even after a blasting operation.

Zhang et al. [23] carried out thermo-mechanical analyses with meso- and macro-scale models and combining the study of fluid dynamics and solidification were able to predict the final track shape and to correlate it with the roughness of the top surface.

The denudation effect across each sequential layer is most likely to cause increment roughness [24]. Thus, to alleviate these undesired effect, layer-to-layer scan rotation was found to be essential to reduce roughness regardless the laser path shape. In addition, spatter particles can be nucleation point of defects impacting on the roughness of successive layers. DePond et al. [25] confirmed the effect of the scanning strategy by carrying out in situ measurements on LPBF manufactured 316L stainless steel by spectral domain optical coherence tomography (SDOCT).

Surface microgeometry of AM parts has been the subject of modelling by several authors. Kaji and Barari [26] proposed a methodology to predict surface roughness, due to the staircase effect, of a FDM part starting from the local surface slope and the layer thickness. Leary [20] proposed a novel methodology to estimate roughness of the Ti6Al4V specimens that distinguishes the relevant and irrelevant surfaces to measure. This could provide indications to optimize product surface quality choosing the appropriate orientation.

Boschetto et al. [27] proposed and validated a model to predict roughness and its dependence on the surface inclination taking into account balling and satellite formation. His approach applied on a surgical fabricated part seems to provide satisfying results.

In this work, the microgeometry of the surface parallel to the building plate of Ti6Al4V alloy specimens produced by LPBF was studied. The AM apparatus used a pulsed laser, and specimens were produced with different combinations of laser power, point distance and time exposure, resulting in energy density in the range between 24.2 and 71.4 J/mm$^2$. Porosity of all the specimens was less than 1%. The surface microgeometry was investigated with a mechanical surface profiler, to acquire the 3D surface profile. Data were elaborated with the approach of the material ratio curve, to determine the representative parameters Mr1, Mr2, Rk, Rpk and Rvk. Moreover, roughness, skewness and kurtosis were determined. All these parameters were calculated on the 3D surface and on some selected linear profiles derived from the intersection of the surface with planes orthogonal to the building plane

and being differently oriented with reference to the scan direction. The aim of the work was:

(a)　to compare the suitability of the 3D and of the 2D approaches to describe the surface microgeometry;

(b)　to investigate the influence of the processing parameters on the surface microgeometry and in particular the effect of point distance and time exposure that have an opposite effect on scan speed and, in turn, on energy density;

(c)　to identify which of the aforementioned parameters are more sensitive to the variations of laser power, time exposure and point distance and are, therefore, more suitable to characterize the surface microgeometry.

## 2. Materials and Methods

The samples were fabricated with a Renishaw® RenAM 500M additive manufacturing system (Renishaw, Wotton-under-Edge, UK), which is a metal powder bed fusion machine operating with a pulsating laser powered by a 500 W ytterbium fibre laser source. The pulsation was determined by two process parameters called point distance (distance between two consecutive points on the powder bed hit by the laser spot) and time of exposure (time interval in which the laser remains active and stationary). The powder was a biomedical grade Ti6Al4V with particles size in the range 15–45 μm.

The working chamber was filled with inert gas to avoid powder oxidation and degradation and kept at 130 °C. Twenty-seven blocks of 20 × 20 × 10 mm were printed by setting three different levels of point distance, time of exposure and laser power. The plan of the three-factor three-level design of experiment (DoE) is summarized in Table 1: they were selected after a preliminary investigation in which parameters were varied in wider ranges to determine the processing window to obtain a porosity lower than 1%. Other main process parameters were kept constant: hatch distance 100 μm, layer thickness 60 μm, path rotation 67°. The specimens were analyzed in as-build conditions without undergoing any post treatment. Further detail about the fabrication process can be found in [28].

**Table 1.** Combinations of process parameters.

| Combination | Point Distance (μm) | Time of Exposure (μs) | Laser Power (W) | Scan Speed (mm/s) | Energy Density (J/mm$^3$) |
|---|---|---|---|---|---|
| 1 | 70 | 40 | 400 | 1750 | 38.1 |
| 2 | 70 | 40 | 450 | 1750 | 42.9 |
| 3 | 70 | 40 | 500 | 1750 | 47.6 |
| 4 | 70 | 50 | 400 | 1400 | 47.6 |
| 5 | 70 | 50 | 450 | 1400 | 53.6 |
| 6 | 70 | 50 | 500 | 1400 | 59.5 |
| 7 | 70 | 60 | 400 | 1167 | 57.1 |
| 8 | 70 | 60 | 450 | 1167 | 64.3 |
| 9 | 70 | 60 | 500 | 1167 | 71.4 |
| 10 | 90 | 40 | 400 | 2250 | 29.6 |
| 11 | 90 | 40 | 450 | 2250 | 33.3 |
| 12 | 90 | 40 | 500 | 2250 | 37.0 |
| 13 | 90 | 50 | 400 | 1800 | 37.0 |
| 14 | 90 | 50 | 450 | 1800 | 41.7 |
| 15 | 90 | 50 | 500 | 1800 | 46.3 |
| 16 | 90 | 60 | 400 | 1500 | 44.4 |
| 17 | 90 | 60 | 450 | 1500 | 50.0 |
| 18 | 90 | 60 | 500 | 1500 | 55.6 |
| 19 | 110 | 40 | 400 | 2750 | 24.2 |
| 20 | 110 | 40 | 450 | 2750 | 27.3 |
| 21 | 110 | 40 | 500 | 2750 | 30.3 |
| 22 | 110 | 50 | 400 | 2200 | 30.3 |
| 23 | 110 | 50 | 450 | 2200 | 34.1 |
| 24 | 110 | 50 | 500 | 2200 | 37.9 |
| 25 | 110 | 60 | 400 | 1833 | 36.4 |

The top surface of specimens was scanned with the KLA-TENCOR P6 (KLA Instruments, Milpitas, CA, USA), which is a mechanical, highly sensitive surface profiler, to

acquire 3D surface profiles of 1 × 1 mm with resolution 2.5 × 3 μm × 0.1 μm (x, y and z directions). The profiler performs three μm-spaced linear scans parallel to X axis with resolution 2.5 μm. Surface characterization was complemented with morphological analyses performed with a JEOL JSM-IT300LV scanning electron microscope (SEM) (JEOL Ltd., Tokyo, Japan). The algorithm proposed by Senin et al. [12] was used for the identification of spatters.

## 3. Results and Discussion

Figure 1 shows, as an example, the SEM images of the top surface of specimens produced with 400 W laser power.

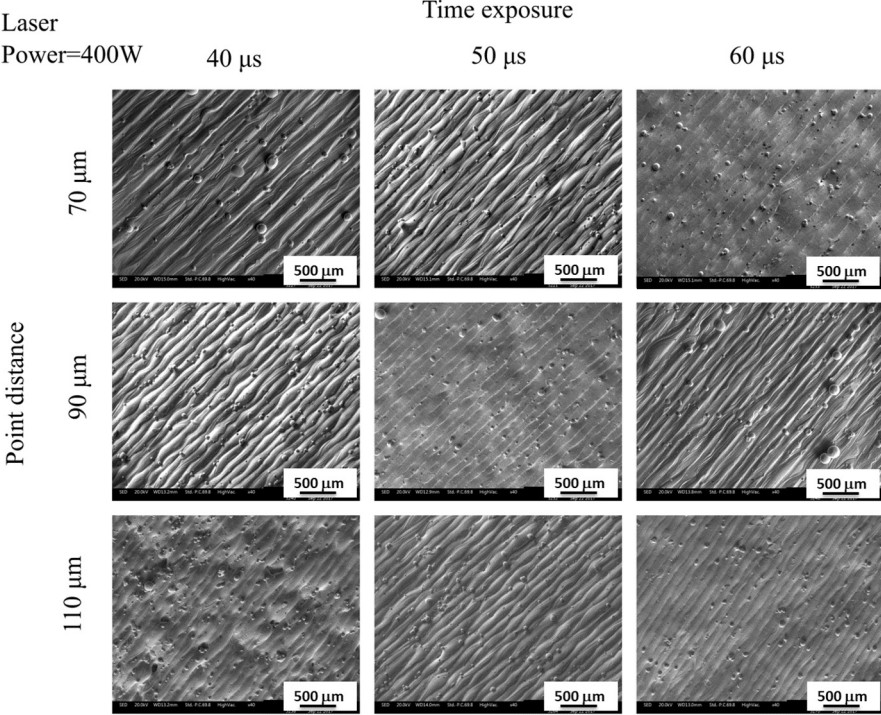

**Figure 1.** Scanning electron microscopy (SEM) micrographs of top surfaces of blocks fabricated with a laser power of 400 W.

The solidified pools have quite regular profiles (indicating a stable melt pool formation) and depict an anisotropic topography. They are oriented in parallel to the scan direction and generate the typical wavy morphology perpendicular to the building direction. However, some of the images highlight a less pronounced waviness parallel to the scan direction (specimen produced with 110 μm point distance and 40 μs time exposure, as an example). Several spatters are also evident on all the specimens. The formation of spatters in LPBF was investigated by Lutter Gunther et al. [29]. They are droplets of molten materials ejected from the melt pool. To verify if spatters may affect the analysis of the surface microgeometry significantly, a procedure was developed to remove them from the profiles recorded by the instrument. In Figure 2, the 3D profile of sample 19 and the results of its elaboration are shown as an example.

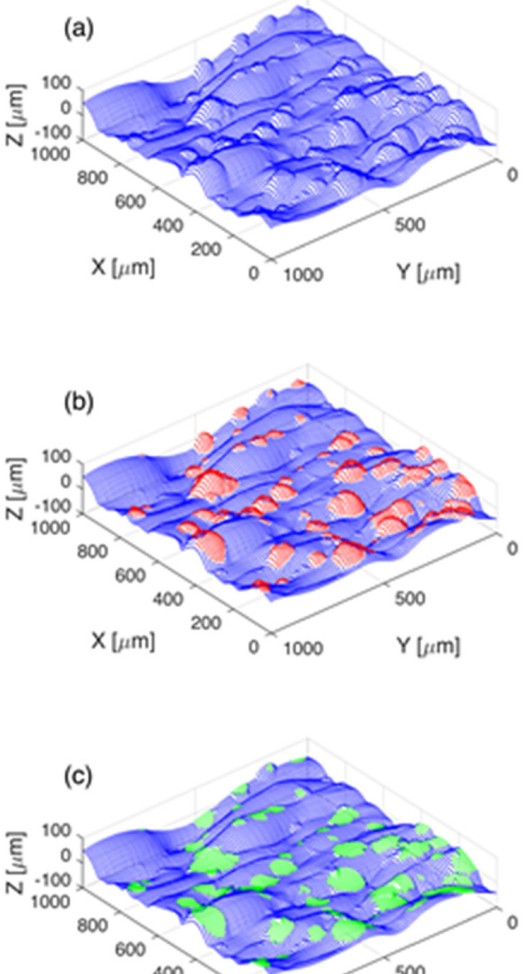

**Figure 2.** Spatter removing procedure; (**a**) original 3D profile normalized, (**b**) identified spatters (red), (**c**) removed spatters, the void regions (green) are filled and reconstructed.

Figure 2a is the 3D profile generated by the instrument. The acquired heights were normalized with respect to the main 3D plane extrapolated with polynomial regression of only one grade (LOESS fitting). It is possible to distinguish the weld tracks, whose direction is perpendicular to the X-axis. Spatters were identified using a procedure in Matlab (MathWorks Inc., Natick, MA, USA) based on the work of Semin et al. [13] that has been already adopted in our previous work [28] (Figure 2b). Spatters identification cannot be based only on the absolute height as those anomalous protrusions lay on a wavy surface. The surface profile was, therefore, filtered with a Gaussian filter with a cut-off frequency set to damp the waviness related to the weld tracks and to make these protrusions detectable. Thus, height-based-identification could be applied. The spatters were removed and replaced by surfaces reconstructed by polynomial regression (Figure 2c). In this way, the effect of spatters was eliminated.

Usually, material ratio curves are obtained from single linear scans following the IS04287 [10]. The material ratio curve is described by the parameters Mr1, Mr2, Rk, Rpk and Rvk shown in Figure 3.

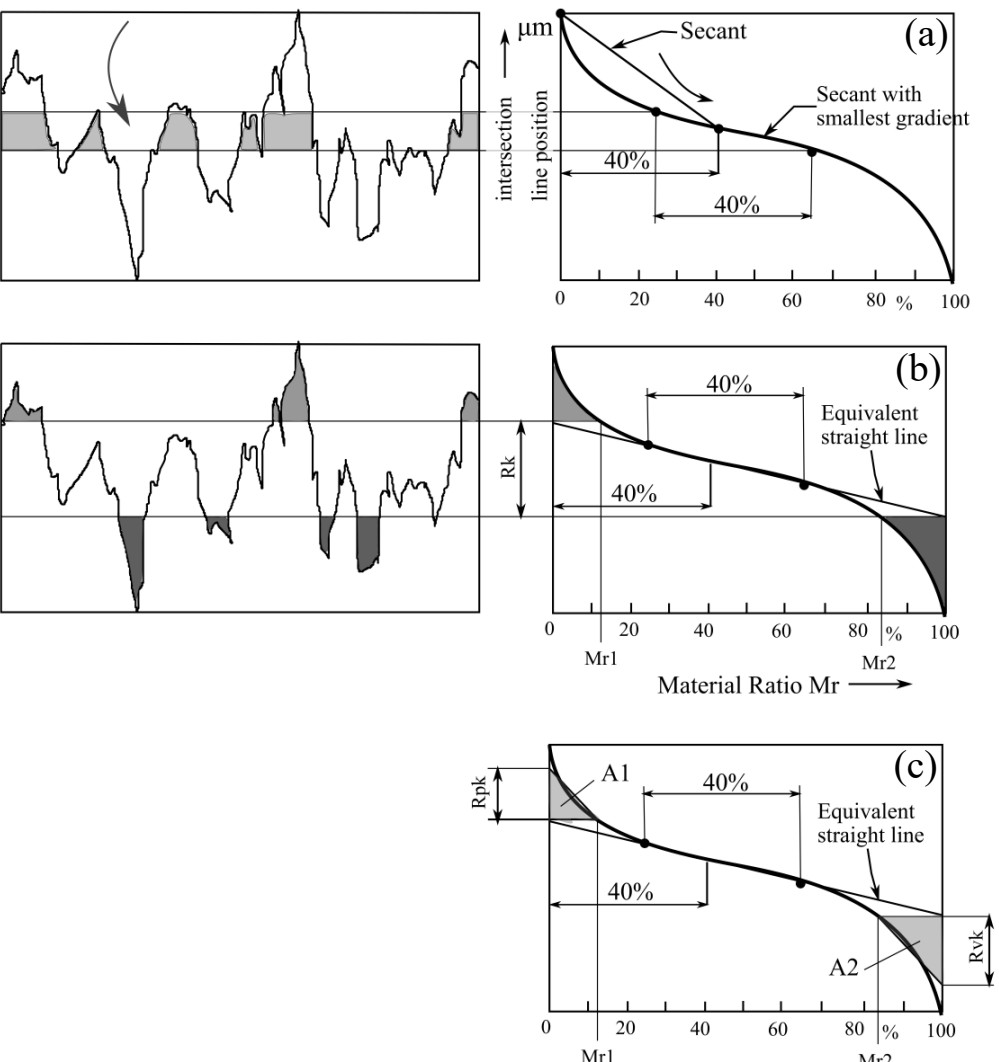

**Figure 3.** Procedure to evaluate surface parameters from a material ratio curve according to the ISO norms. Rk is the height range corresponding to the equivalent straight line found by shifting a secant, having 40% of width, up to the smallest inclination (**a**). Mr1 and (100−Mr2) represent the portions of the surface covered by peaks and valleys, respectively, and define the two black regions in (**b**). Rpk and Rvk are the height of the triangles having the same area of those regions (**c**).

The procedure for their determination is codified by the ISO4288 standard [30]. The equivalent straight line is found by shifting a secant, having 40% of width, up to the smallest inclination (Figure 3a). Rk is the height range corresponding to the equivalent line. Mr1 and (100-Mr2) represent the portions of the surface covered by peaks and valleys, respectively, and define the two black regions in Figure 3b. Rpk and Rvk are the height of the triangles having the same area of those regions (Figure 3c). In tribology, Mr2 represents the load-bearing surface.

In Figure 4 the same concept is applied for a 3D surface.

Three-dimensional material ratio curves are determined according to the procedure sketched in Figure 4. Accordingly, a plane parallel to XY plane is positioned at the highest surface peak (Figure 4a) and then move down towards the minimum point. At each height Z the amount of material lying above that plane is recorded and the plot shown in Figure 4b is generated.

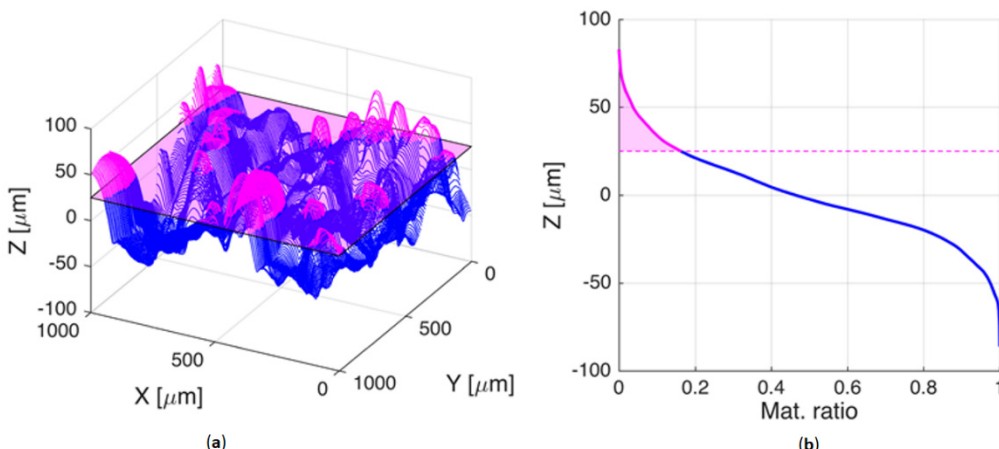

**Figure 4.** Procedure for evaluating material ratio curve in 3D: (**a**) the XY plane and (**b**) the resulting curve.

In Figure 5a, the 3D material ratio curve and the 2D ones relevant to 300 parallel scans along the X axis (perpendicular to the scan direction) of sample 19 are compared.

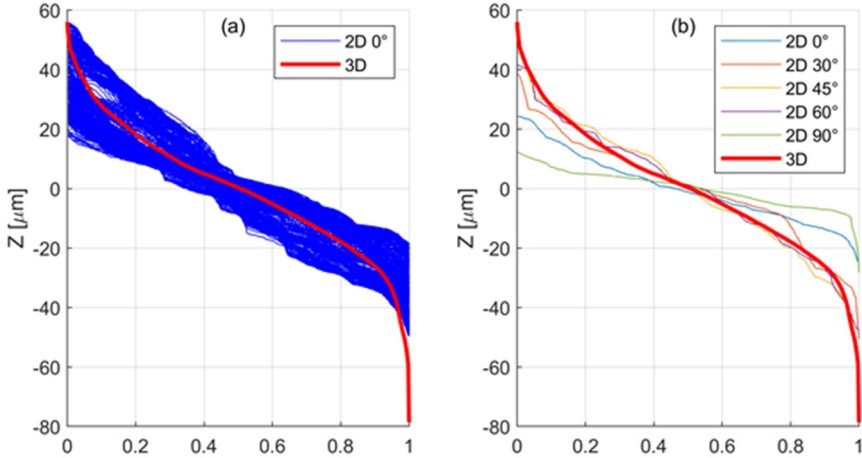

**Figure 5.** Comparison between material ratio curves evaluated in 2D and in 3D (red); (**a**) 3D curve compared to the band generated by 2D curves and (**b**) 3D curve compared with five 2D curves collected along different directions.

It is worth noting in Figure 5a that 2D curves display a large dispersion, their trend is nevertheless well described by the 3D curve, which lies in the middle of the envelope of the band generated by the 2D curves. In addition, while the peak height of the red curve coincides with the maximum value assumed by the blue ones, the 3D approach seems to better reveal the lowest peak of the height profile (i.e., the deep valleys) with respect to the convectional 2D approach. This could be explained as follows. When each single 2D profile acquired is processed and normalized with respect to its mean line, the entire surface morphology is not taken into account, and this is of particular relevance in the case of anisotropic surfaces, such as those investigated in the present work. In Figure 5b, the 3D curve is compared with five curves collected along different directions, ranging from that parallel to the X axis (0° in the label) to that parallel to the Y axis (90° in the label). As might be expected, the flattest profile is parallel to the scan direction. In contrast to what it might be expected, the curve perpendicular to the scan direction shows neither the steeper trend nor the highest peaks and deepest valley among the 2D curves. The 3D curve, which better represents the anisotropic microgeometry of the surface, exhibits a shape more similar to the 2D ones collected along intermediate directions and allows us to better characterize the peaks and the valleys. When a 2D linear profile scan is performed, the profile must be

normalized with respect to the average line. Welding track is not so perfectly homogeneous along its length, so it is reasonable to expect a variation of height even parallel to the weld track.

In Figure 6, the impact of spatters on the material ratio curves is shown for samples 1 and 19. After eliminating the spatters, a slight downward shift is visible, but the main inclination does not vary so much.

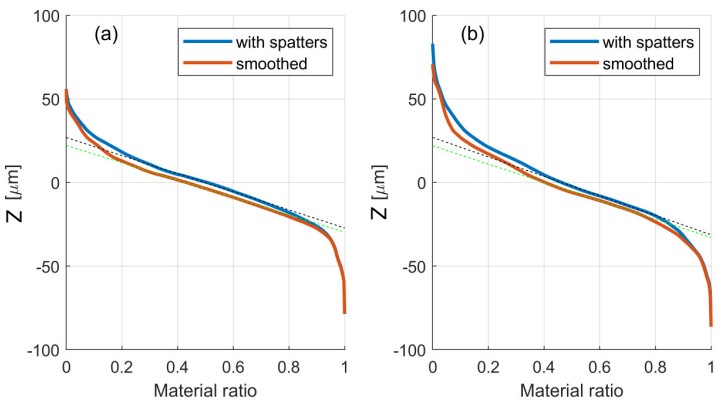

**Figure 6.** Material ratio curves from samples (**a**) 1 and (**b**) 19. Effect of removing spatters. The dashed line indicates the equivalent straight line.

In accordance to the DoE approach, the main effects of point distance, time exposure and laser power on the roughness parameters calculated from the 3D curve and from five 2D curves measured at different orientations is presented below. Figure 7 shows the effect on Mr1 and Mr2. Each datum is the average of the results of the experiments that have been carried out with the value of the abscissa which the data correspond to; the same approach was used to draw the figures that follow.

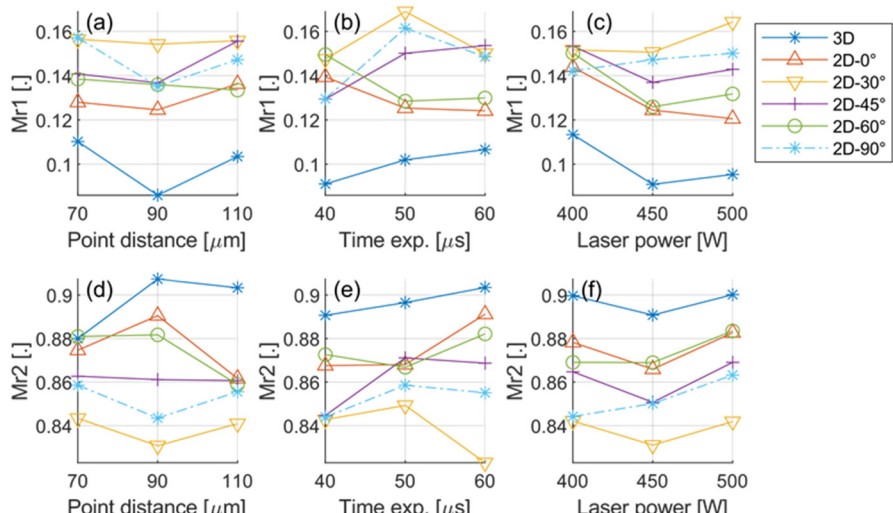

**Figure 7.** Influence of process parameters on (**a**–**c**) Mr1 and (**d**–**f**) Mr2. Effect of (**a**,**d**) point distance, (**b**,**e**) time exposure (exp.) and (**c**,**f**) laser power.

The first result is that Mr1 of the 3D curve is slightly lower than that of the 2D counterparts, while Mr2 is slightly higher. This means that the 2D characterization slightly overestimates the portions of the surface covered by peaks and valleys. As far as the process parameters are concerned, their effect is almost negligible, essentially for two reasons: (i) only in a few cases is there a monotonic trend; (ii) the difference between the highest and the smallest values is very small (maximum 0.02%).

Figure 8 shows the effect of the process parameters on Rk, Rpk and Rvk.

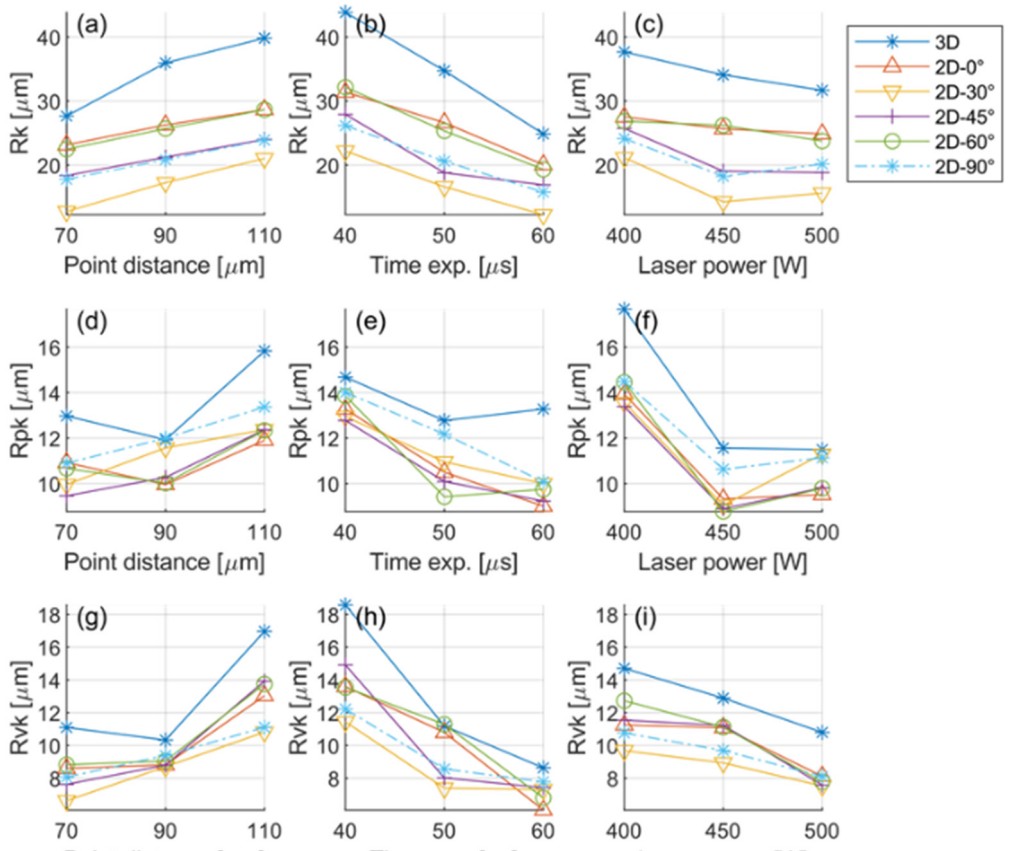

**Figure 8.** Influence of process parameters on (**a-c**) Rk, (**d-f**) Rpk and (**g-i**) Rvk. Effect of (**a,d,g**) point distance, (**b,e,h**) time exposure (exp.) and (**c,f,i**) laser power.

It can be noted that the 3D characterization gives higher values of the three parameters, in particular a higher Rk than the 2D characterization, confirming that 2D analysis underestimates the irregularity of the surface microgeometry. The effect of the process parameters is evident. Rk increases with increasing point distance and decreasing time exposure and laser power, namely, with increasing scan speed and decreasing energy density (given by the ratio between the laser power and the scan speed). The effect on Rpk and Rvk is less sharp, even if the figure clearly indicates that the increase in the laser power and the decrease in the scan speed result in lower peaks and shallower valleys.

Figures 9 and 10 show the effect of the process parameters on Rq and Ra of the 2D profiles measured along different directions and on Sq and Sa determined form the 3D curves, respectively.

Ra, Rq, Sa and Sq are significantly affected by the process parameters: they increase with increasing point distance and decreasing time exposure and laser power, i.e., increasing scan speed and decreasing the energy density. Sa and Sq are larger than Ra and Rq, respectively. The direction does not have a systematic effect on Ra and Rq.

Skewness is slightly positive, indicating that the largest percentage of the profile lies below the mean line without a significant effect of the processing parameters. Two- and three-dimensional skewness values are very similar. Two-dimensional kurtosis values are less than 3, indicating a uniform/normal distribution of peaks and valleys without an effect of the processing parameters. The 3D kurtosis is higher (between 3 and 4.5), indicating that the 3D profile is characterized by slightly sharper peaks and valleys than the 2D one.

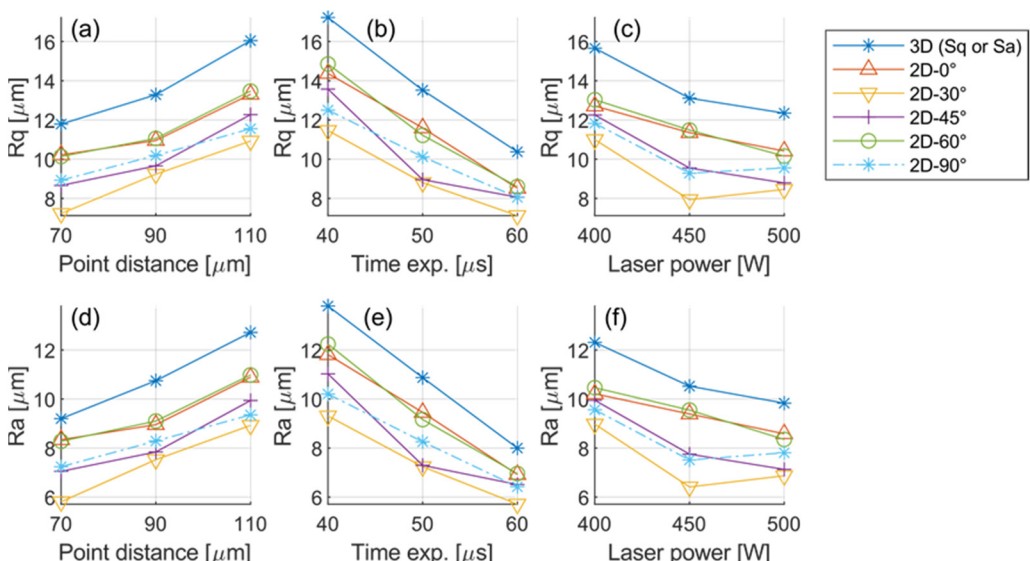

**Figure 9.** Influence of process parameters on (**a**–**c**) Rq and (**d**–**f**) Ra (Sq and Sa for 3D data) evaluated along x and y axis. Effect of (**a**,**d**) point distance, (**b**,**e**) time exposure (exp.) and (**c**,**f**) laser power.

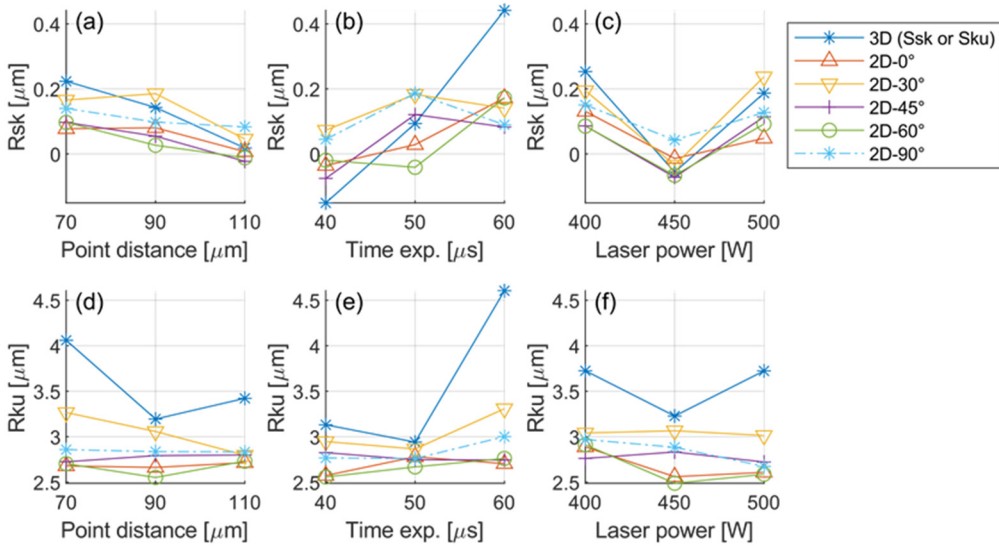

**Figure 10.** Influence of process parameters on (**a**–**c**) skewness Rsk and (**d**–**f**) kurtosis Rku. Effect of (**a**,**d**) point distance, (**b**,**e**) time exposure (exp.) and (**c**,**f**) laser power.

In conclusion, it can be stated that the processing parameters exert a significant effect on the roughness parameters Ra, Rq, Sa, Sq and Rk. Within the investigated ranges, the increase in laser power and time exposure and the decrease in point distance decrease roughness and tend to decrease the peak height (Rpk) and the valley depth (Rvk). This is due to the effect of the temperature of the melt pool on the aspect ratio that decreases with increasing energy density [15]. The distribution of the profile is not affected significantly by the investigated processing parameters.

## 4. Conclusions

In this work the surface microgeometry of Ti6Al4V specimens produced by a pulsed LPBF process was investigated. The influence of the main process parameters: laser power, point distance and time exposure was studied. A 3D characterization was carried out with a mechanical surface profiler, to acquire 3D surface profiles with resolution 2.5 × 3 μm. Collected data were elaborated to reconstruct the surface and to determine both the 3D

and the 2D material ratio curves along different directions. A special algorithm was implemented to eliminate spatters from the reconstructed surface to evaluate their effect on the microgeometry.

Due to the pronounced texture, the 3D approach gives a better representation of the surface microgeometry than the 2D approach whose results are strongly dependent on the direction of the measure. The 3D material ratio curve gives slightly lower Mr1 and higher Mr2, Rk, Rpk and Rvk than the 2D curves. Concerning roughness, Ra and Rq are greater in the 3D analysis than in the 2D one, skewness is the same and kurtosis increases from <3 in 2D to >3 in 3D.

The effect of the spatters is poor, they only cause a slightly greater Rpk in the material ratio curve.

The processing parameters have a significant effect on the surface microgeometry. All the parameters that signify roughness, Ra, Rq, Sa, Sq, Rk increase on increasing point distance and decreasing time exposure and laser power. Within the investigated ranges ($27.3–71.2 \, J/mm^3$), an increase in energy density reduces the surface roughness, regardless of how it is attained (increase in the laser power and time exposure, decrease of point distance), while the distribution of the profile, as represented by skewness and kurtosis, is not significantly affected.

The results of this work will be verified by investigating the microgeometry of surfaces having a different orientation with respect to the build plate.

**Author Contributions:** Conceptualization, V.F. and M.B.; methodology, S.A. and A.M.; validation, G.Z. and M.B.; formal analysis, V.F.; investigation, S.A., E.I. and V.L.; resources, G.Z.; data curation, S.A. and V.L.; writing—original draft preparation, S.A. and A.M.; writing—review and editing, V.F., G.Z. and M.B.; funding acquisition, G.Z. All authors have read and agreed to the published version of the manuscript.

**Funding:** This research was supported by the Autonomous Province of Trento [FAMAC project, Reg. delib. 520/2018].

**Data Availability Statement:** The data presented in this study are available on request from the corresponding author. The data are not publicly available due to confidentiality.

**Conflicts of Interest:** The authors declare no conflict of interest. The funders had no role in the design of the study; in the collection, analyses, or interpretation of data; in the writing of the manuscript, or in the decision to publish the results.

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
