# Peer review of "Effect of Process Parameters on the Surface Microgeometry of a Ti6Al4V Alloy Manufactured by Laser Powder Bed Fusion: 3D vs. 2D Characterization"

_metals, doi:10.3390/met12010106_

Round 1

Reviewer 1 Report

For this publication, the surface microgeometry of LPBF Ti6Al4V was measured using a sensitive surface profiler and subsequently evaluated in 2D and 3D. For this purpose, 27 samples were additively fabricated, 25 different process parameter combinations were investigated and the results of individual samples were presented in more detail. The introduction provides a sufficient overview of previously published results.

The results are adequately presented, however, a more in-depth discussion of the results as well as an outlook would be recommended. In addition, the basic motivation of the work should be elaborated more. The following points could be considered:

  • Why was the surface parallel to the build plate chosen? Normally, the proportion of this surface orientation of a component after the printing process is only small (unless a sheet/ thin plate is printed). Furthermore, this surface orientation will hardly dominate the component properties, since the roughness is usually greater in other orientations. During the process, this surface is of course relevant, since surface irregularities as described in the text can e.g. lead to a collision of the recoating blade with the components. However, the surface irregularities that must occur for this damage are much larger than the effects/standard surfaces studied in this publication.
  • Further, many roughness parameters are evaluated and listed in the publication. However, there is not enough detail on what the effect of the parameters is: what parameters should be used to describe the surface? (line306: “in conclusion, it can be stated that the processing parameters exert a significant effect on the roughness parameters Ra, Rq, Sa, Sq and Rk”. and line 313: “The distribution of the profile is not affected significantly by the investigated processing parameters”). Does this mean, as a conclusion, that the material ratio curve is hardly meaningful in practice and therefore does not need to be investigated? (Although the material ratio curve is a large part in the publication)
  • What can be found out with the parameters determined? Can process parameters be optimized with it and if so, how? Does the surface area affect the porosity of the samples? Can it be used to explain volume errors that occur during the process?
  • In the introduction it was mentioned that according to reference [17] kurtosis (Sku) and skewness (Ssk) are better suited to describe LPBF surfaces compared to Sa and Sq. Is there a reason why these parameters were not investigated here, although they were specifically mentioned in the introduction?
  • Line 311-313: "Hence, within the investigated range of process parameters the weld track appears to remain stable up to energy density values of ca. 70 J/mm3." - Can the stability of melt tracks be determined via the roughness parameters? The sentence comes as a relative surprise at this point.
  • Can the results obtained be compared with values from the literature?

The evaluations of the results lack information on reproducibility/error estimation:

  • In the publication, an area of 1mmx1mm is evaluated from each sample. Is there any experience to what extent this area is sufficient? Or is a larger area recommended? Are the measurements reproducible? Were several identical samples measured? If the size of the area plays a role: Which roughness values are most influenced by the area?

Minor comments:

  • A table in the appendix with the values for all samples would be helpful.
  • The following information should be added in the method section:
    line 172: where was the powder purchased?
    line 173/174: Which inert gas was used? Was the working chamber kept at 130°C or was the baseplate set to 130°C?
    line 183: What is the resolution in z-axis? (x,y: 2.5µmx3µm. z=?)
  • line 13/317: no colon (“:”) -> the influence of the main process parameters Laser Power, …. was investigated
  • please check the uniformity of spelling:
    • Ti6Al4V (e.g. title) vs Ti-6Al-4V (line 39)
    • Material ratio (line 17) or material ratio (line 84)
  • please use current technical terminology: LPBF (laser powder bed fusion) instead of SLM (selective laser melting)
  • line 254/255: Do you have an explanation for your result: “In contrast to what it might be expected, the curve perpendicular to the scan direction shows neither the steeper trend nor the highest peaks and deepest valley among the 2D curves.”
  • Figure 7 to 10: Please add the missing process parameters in the caption (e.g. laser Power, ….). It is not obvious, which specimens' results are depicted.
  • Figure 6: really nice result; Do you know how the spatter affects Sa/Ra and Sq/Rq?

Reviewer 2 Report

The manuscript entitled “Effect of Process Parameters on the Surface Microgeometry of a  Ti6Al4V Alloy Manufactured by Powder Bed Fusion: 3D vs 2D  Characterization” dealing with PBF has been reviewed. The paper is well written and needs the following improvements.

  1. Add a short note about the results to the abstract.
  2. Authors are encouraged to provide a statement on how they selected the combination of process parameters presented in Table 1.
  3. The introduction needs to be updated with the following references.

  • Kumar, M., G.J. Gibbons, A. Das, I. Manna, D. Tanner, and H.R. Kotadia, Additive manufacturing of aluminium alloy 2024 by laser powder bed fusion: microstructural evolution, defects and mechanical properties. , 2021.
  • Guan, J., Q. Wang, C. Chen, and J. Xiao, Forming feasibility and interface microstructure of Al/Cu bimetallic structure fabricated by laser powder bed fusion. , 2021.
  • Mezghani, A., A.R. Nassar, C.J. Dickman, E. Valdes, and R. Alvarado, Laser powder bed fusion additive manufacturing of copper wicking structures: fabrication and capillary characterization. , 2021.
  • Gonzalez Alvarez, A., P.L. Evans, L. Dovgalski, and I. Goldsmith, Design, additive manufacture and clinical application of a patient-specific titanium implant to anatomically reconstruct a large chest wall defect. , 2021.
  • Shaikh, M.Q., S. Graziosi, and S.V. Atre, Supportless printing of lattice structures by metal fused filament fabrication (MF3) of Ti-6Al-4V: design and analysis. , 2021.

  1. The scale bars in Figure 1 are not readable. Please improve them.

  1. Additive manufacturing has many advantages over the conventional manufacturing method which can be highlighted in your paper. Please read the following article and add to the introduction to show the experimental application of additive manufacturing and the advantage of this process over conventional manufacturing like machining.

Additive manufacturing a powerful tool for the aerospace industry.

  1. Add more explanation about figure five in the text.
  2. Future work should be added to the paper.
  3. More details about figures 8 to 10 are required. This robust the paper and the quality of the discussions.

Reviewer 3 Report

It is a great job and a very readable and enjoyable article, but, if I have understood correctly, I don't see any practical value in the research conducted. 

The article makes reference to spatters that are not going to be on the surface, they will be in the bulk of the part and they will be hidden by the next layer. 

Therefore, one of the most important factors in the roughness patterns of the LPBF process, which is the layer thickness, is not taken into account.

In the opinion of this reviewer, if I have understood correctly, roughness is being evaluated with parameters from the bulk of the part and not from the borders, which is where it have to be measured.  If all the measurements are repeated with borders parameters in the other surfaces of the cube it will be a good and interesting article.

Round 2

Reviewer 1 Report

Thank you for incorporating the comments. Regarding roughness, my personal experience has been different - usually roughness in the upskin region is much lower than in the downskin region, but it is interesting to read that you will extend your model to other orientations, which is very welcome. It is up to you to decide whether you want to provide the table with the values. If you do not feel the table is helpful to the reader, there is no need to include the table.

Author Response

Thank you for appreciating our revised paper

Reviewer 2 Report

The paper is publishable. 

Author Response

(The authors gave the same response as above.)

Reviewer 3 Report

I still don't understand why an up-skin surface has been used to evaluate a 2D/3D surface roughness and its application in a real part. In LPBF as built real parts the most influential parameter on roughness is layer thickness, despite being constant, and authors has chosen to evaluate part roughness the up-skin surface of that is not a normal surface in 3D real parts. To evaluate the roughness, it is necessary to choose the lateral surfaces where it is implicit layer thickness.

As far as I am concerned, I cannot accept the paper without a practical explanation of the purpose of this study on real parts.

Author Response

I agree with the Reviewer about the importance of characterizing the lateral surface, too. I also think that the characterization should involve the effect of the orientation of the surface, for a comprehensive investigation of the surface roughness. This is the subject of future work. We have chosen the top surface, parallel to the build plate, since it represents the single layer, and its microgeometry will surely affect the final structure of the parts (bad roughness may lead to the formation of large pores, just as an example). I also agree about the importance of the layer thickness, but this parameter was not considered since it was previously optimized by the Company partner. Moreover, our previous works on precious metal showed that roughness is the worst on the surface parallel to the building plate. Finally, the work we carried out allowed us to validate the algorithm for the processing of spatters and to define the experimental approach for the further study. I hope that this explanation might convince the Reviewer about the interest of our work.

Round 3

Reviewer 3 Report

I am very sorry but I need to find the applicability and industrial interest in this type of article. If the same research is done on the lateral surfaces of the part (cube) it would be a suitable article for publication and would be accepted by the present reviewer.

I have been working for years in researching with aeronautical companies in high TRLs for powder bed fusion process and we have never measured the roughness on the surface that you have measured in because it doesn´t have practical application.

On the other hand, I also consider myself an expert in 2D/3D roughness measurement, and I think that the work done with roughness is quite good. Is why for me is difficult to regetc this article but I can´t accept it, sorry.